# Maternal Fructose Intake Causes Developmental Reprogramming of Hepatic Mitochondrial Catalytic Activity and Lipid Metabolism in Weanling and Young Adult Offspring

**DOI:** 10.3390/ijms23020999

**Published:** 2022-01-17

**Authors:** Erin Vanessa LaRae Smith, Rebecca Maree Dyson, Christina M. G. Vanderboor, Ousseynou Sarr, Jane Anderson, Mary J. Berry, Timothy R. H. Regnault, Lifeng Peng, Clint Gray

**Affiliations:** 1Department of Paediatrics and Child Health, University of Otago, Wellington 6021, New Zealand; erin.smith@postgrad.otago.ac.nz (E.V.L.S.); becs.dyson@otago.ac.nz (R.M.D.); max.berry@otago.ac.nz (M.J.B.); 2Centre for Translational Research, University of Otago, Wellington 6021, New Zealand; 3Department of Physiology and Pharmacology, Western University, London, ON N6A 3K7, Canada; cgodin2@uwo.ca (C.M.G.V.); osarr@uwo.ca (O.S.); 4Department of Pathology & Molecular Medicine, University of Otago, Wellington 6021, New Zealand; jane.anderson@otago.ac.nz; 5Department of Obstetrics and Gynaecology, Children’s Health Research Institute, Western University, London, ON N6A 3K7, Canada; tim.regnault@uwo.ca; 6Centre for Biodiscovery and School of Biological Sciences, Victoria University of Wellington, Wellington 6021, New Zealand

**Keywords:** maternal fructose, excess fructose intake, developmental programming, hepatic metabolism, mitochondrial metabolism, mitochondrial function

## Abstract

Excess dietary fructose is a major public health concern, yet little is known about its influence on offspring development and later-life disease when consumed in excess during pregnancy. To determine whether increased maternal fructose intake could have long-term consequences on offspring health, we investigated the effects of 10% *w*/*v* fructose water intake during preconception and pregnancy in guinea pigs. Female Dunkin Hartley guinea pigs were fed a control diet (CD) or fructose diet (FD; providing 16% of total daily caloric intake) ad libitum 60 days prior to mating and throughout gestation. Dietary interventions ceased at day of delivery. Offspring were culled at day 21 (D21) (weaning) and at 4 months (4 M) (young adult). Fetal exposure to excess maternal fructose intake significantly increased male and female triglycerides at D21 and 4 M and circulating palmitoleic acid and total omega-7 through day 0 (D0) to 4 M. Proteomic and functional analysis of significantly differentially expressed proteins revealed that FD offspring (D21 and 4 M) had significantly increased mitochondrial metabolic activities of β-oxidation, electron transport chain (ETC) and oxidative phosphorylation and reactive oxygen species production compared to the CD offspring. Western blotting analysis of both FD offspring validated the increased protein abundances of mitochondrial ETC complex II and IV, SREBP-1c and FAS, whereas VDAC1 expression was higher at D21 but lower at 4 M. We provide evidence demonstrating offspring programmed hepatic mitochondrial metabolism and de novo lipogenesis following excess maternal fructose exposure. These underlying asymptomatic programmed pathways may lead to a predisposition to metabolic dysfunction later in life.

## 1. Introduction

Fructose has become a major public health concern due to the associated increase of metabolic diseases such as; dyslipidemia, hyperlipidemia, insulin resistance, visceral adiposity, obesity, diabetes, high blood pressure, cardiovascular disease and non-alcoholic fatty liver disease (NAFLD) [1,2,3,4,5,6,7]. When considering the impact of fructose on the prevalence of metabolic disease, several factors appear clear. In the presence of readily available hepatic glucose and glycogen stores, key metabolic pathways of fructose metabolism become unregulated and excess fructose is channeled to lipogenesis. Consequently, the chronic metabolic alteration caused by excess fructose consumption initiates an adaptive series of responses in signaling pathways. These adaptive responses to an abnormal hepatic de novo lipogenesis may contribute to altered mitochondrial catalytic activity and function and metabolic dysregulation [1,5,8]. However, no studies have examined the mechanistic impact of increased fructose intake during pregnancy and subsequent adverse effects on offspring mitochondrial metabolism.

Research has shown that suboptimal maternal nutrition and subsequent nutrient availability supplied to the blastocyst, fetus, neonate and developing infant, and the timing of nutritional ‘insults’ are important for determining later-life health and non-communicable diseases (NCDs). To better understand offspring predisposition to metabolic disease following excess maternal fructose exposure in utero, it is important to understand the molecular changes associated with fructose intake. During fetal development, the liver undergoes many changes in structure and function that can influence the rate of nutrient absorption, metabolism and redistribution of metabolites into circulation. It is therefore important to determine how the vertical transmission of deleterious metabolic effects influences de novo lipogenesis, fatty acid β-oxidation, mitochondrial function and subsequent metabolic reprogramming in offspring.

Developmental programming of mitochondrial metabolism presents a key intracellular candidate underlying many phenotypes that are observed within the Developmental Origins of Health and Disease (DOHaD) paradigm [9]. In a rat model of maternal high fat feeding, Pileggi et al. reported differences in mitochondrial catalytic activity in the skeletal muscle of adult offspring in response to a high fat diet [10]. They showed reduced expression of nuclear respiratory factor-1 (NRF1), mitochondrial transcription factor-A (mtTFA) and reduction in genes involved in mitochondrial catalytic activity in the electron transport chain (ETC) complexes I and III [10]. Animal studies have also shown links between programmed mitochondrial function in offspring and increased reactive oxygen species (ROS) [11,12,13,14,15], which may be associated with decreased ATP production [16]. Although human studies are limited, one study by Leary et al. showed that oocytes from obese mothers had reduced mitochondrial function and impaired oxidative phosphorylation [17]. Moreover, mitochondrial DNA content has been observed to be reduced in placentas from women exposed to environmental stimuli such as obesity [18,19], stress [18,20], smoking [18,19] and pollutants [21,22,23]. Clearly current research indicates an important role for maternal nutritional status and extrinsic environmental factors in the developmental programming of offspring mitochondrial function. In the context of maternal fructose intake, there are no published reports examining the impact of excess fructose intake during pregnancy or the adverse effects on offspring hepatic mitochondrial function. In the current study, we aimed to investigate the effects of excess maternal fructose intake upon hepatic de novo lipogenesis and mitochondrial function in male and female offspring from birth to adulthood.

## 2. Materials and Methods

### 2.1. Ethics

All studies were conducted with the approval of the University of Otago, Wellington, Animal Ethics Committee (AEC: 7–15) and performed in line with the *Guide for the Care and Use of Laboratory Animals*, 8th Edition, and the National Animal Ethics Advisory Commission, New Zealand [24]. Results are reported according to the ARRIVE guidelines [25].

### 2.2. Experimental Diets

All guinea pig dams, and offspring received standard guinea pig pellets (guinea pig pellets; Sharpes Stock Feeds, Carterton, NZ), hay ~10 g and fresh vegetables (half a piece of fresh silver beet and ~18 g carrots daily). Millipore filtered water supplemented with 1.2 g/L of ascorbic acid was available ad libitum. The dams’ water was additionally supplemented with 10% fructose water (Fructose group) from 12 weeks to birth (~69 days of pregnancy). Dam caloric intake and weight gain were recorded and published [7].

### 2.3. Animal Model

All animals were housed in polypropylene cages (Tecniplast, Lane Cove, Australia) lined with pine wood shavings under a 12 h day/night light cycle within a temperature (18–23 °C)- and humidity-controlled facility. In brief, the maternal groups included control diet (CD) dams (*n* = 10) and fructose diet (FD) dams (*n* = 9). At 12 weeks, the dams were matched for weight and randomly assigned to CD and FD groups. At 6 months, they were housed with a non-lineage boar during estrous for 72 h. Following mating, dams were returned to individual cages and continued their experimental diets throughout gestation. Experimental dietary interventions were ceased and all dams and offspring maintained on a control diet following birth. Pups were sexed and all litters were maintained at four pups (*n* = 2 for males, *n* = 2 for females).

One sibling of each sex was assigned to either the weanling (D21; all groups *n* = 7) or the adult (4 M; CD males (*n* = 7); FD male (*n* = 9); CD female (*n* = 7); or FD female (*n* = 6)) group. Oral glucose tolerance tests (OGTTs) were performed at day 21, 2 months and 4 months. All offspring were weaned at D21 and maintained on a controlled diet of standard guinea pig pellets, hay, fresh silver beet and carrots and Millipore filtered water supplemented with 1.2 g/L of ascorbic acid until 4 M of age. Following the final OGTT at 4 M, offspring underwent body composition assessment using a dual-energy, X-ray absorptiometry (DXA) scan then euthanized, so blood and tissue samples could be collected.

### 2.4. Determination of Body Composition

Dual-energy X-ray absorptiometry (DXA) (Hologic Horizon DXA System, Mississauga, ON, Canada) scans were performed on offspring at 4 M using the Hologic Horizon DXA System (Hologic, Mississauga, ON, Canada). All animals were fasted for 14 h overnight with ad libitum access to water. Prior to scanning, guinea pigs were weighed and anesthetised by an intramuscular injection mixture of ketamine (40 mg/kg, PhoenixPharm, Auckland, New Zealand) and medetomidine (0.5 mg/kg, Domitor^®^, Pfizer Animal Health, Auckland, New Zealand).

### 2.5. Offspring Oral Glucose Tolerance Tests and Blood Glucose Analysis

OGTTs were performed on offspring using our established protocol [7]. Animals were fasted for 14 h overnight. Following a baseline blood collection, an oral dextrose load (1000 mg/kg) was administered with serial blood collections for glucose and insulin concentrations at 15, 30, 45, 60, 75, 90, 120 and 180 min. The Matsuda–DeFronzo Insulin Sensitivity Index (M–ISI) was used to evaluate whole-body insulin sensitivity [26,27].

### 2.6. Offspring Whole Blood Free Fatty Acids Analysis

At 2 M and 4 M approximately 40 μL of blood was collected from the auricular ear vein and placed on a Dried Blood Spot (DBS) PUFAcoat™ card (University of Adelaide, Adelaide, Australia), dried and stored at room temperature until further analysis. As previously described, offspring whole blood measurement for free fatty acids were performed using a modified direct transesterification method [7,28].

### 2.7. Offspring Visceral Adipose and Liver Tissues Collection

Immediately following OGTTs at D21 and 4 M, offspring were humanely euthanised, visceral adipose and liver tissues were collected and weighed (Appendix A). Tissues were snap-frozen in liquid nitrogen and stored at −80 °C for proteomic and metabolic analyses or fixed in 10% formalin phosphate buffer for microscopic examination.

### 2.8. Fixation of Visceral Adipose and Liver Tissues with Formalin

Visceral adipose tissue was dissected and fixed in 10% formalin for 24 h before storage in 0.1 M phosphate buffer. Following paraffin fixing, 4 μm thick tissue sections were cut using a Leica RM2235 microtome (Leica Biosystems, Auckland, New Zealand).

Right lower dorsal lobes of livers were dissected and snap-frozen in liquid nitrogen and stored at −80 °C. Following embedding, 6 μm thick tissue sections were cut and placed onto Superfrost plus^®^ slides. Two serial sections were taken from 2 tissue depths approximately 80 μm apart. Microscopy was performed on an Olympus BX51 microscope with images captured using an Olympus DP20 mounted camera (Olympus, Wellington, New Zealand). Five fields of view at 100×, 200×, or 400× magnification [29,30] were captured from each section (3 sections/sample).

### 2.9. Plasma Metabolite Analysis

Offspring plasma were analysed by Cobas c311 autoanalyzer (Roche, Basel, Switzerland) according to manufacturer instructions. Offspring plasma insulin and post-mortem leptin concentrations were determined by a guinea pig quantitative competitive immunoassay kit (ELISA) (Abclonal, Woburn, MA, USA) according to the manufacturer’s instructions.

### 2.10. Determination of Hepatic Triglycerides

Offspring hepatic triglycerides were determined by enzymatic colorimetric assays (Triglyceride Colorimetric assay kit, Cayman Chemical, Ann Arbor, MI, USA) according to manufacturer’s instructions.

### 2.11. Protein Extraction from Liver Tissues

Fifty milligrams of each of the liver tissues from D21 and 4 M offspring in duplicate were aliquoted and homogenised in lysis buffer (40 mM Tris-HCl pH 7.4, 7 M urea, 2 M thiourea, 4% 3-[(3-cholamidopropyl) dimethylammonio]-1-propanesulfonate and protease inhibitor cocktail P8340 (Sigma-Aldrich, Oakville, ON, Canada) on ice with a 1 mL Dounce homogeniser (Sigma-Aldrich). The homogenate was vortexed, centrifuged and the supernatants (protein extract) were collected. Protein concentration was measured by Bradford Assay (Bio-Rad Protein Assay) in a 96-well plate (Corning, TX, USA).

### 2.12. Protein Precipitation Digestion and Peptide Purification

Using ProteoExtract Protein Precipitation Kit (Merck Ltd., Auckland, New Zealand) 30 µg of proteins were precipitated according to the manufacturer’s instructions and resuspended in 100 mM Tris-HCl, pH 8.8, and 8 M urea and then digested with trypsin (v5111, Promega, Madison, WI, USA). Purification of resulting tryptic peptides was accomplished with OMIX C18 Ziptips (Agilent Technologies Inc., Santa Clara, CA, USA) according to the manufacturer’s protocol. Peptides were eluted in 20 μL of 0.15% formic acid (FA) in 50% acetonitrile (ACN) and 20 μL of 0.15% FA in 70% ACN, respectively, which were subsequently combined and dried down to ~10 μL at 35 °C using SpeedVac (Labconco, Kansas, MO, USA) in a 0.5 mL Protein LoBind^®^ tube (Eppendorf, Hamburg, Germany). The peptides were placed in 0.1% FA in water to make a final solution volume of 140 μL, vortexed and centrifuged at room temperature and then transferred to a glass vial with a glass inserts (ThermoFisher Scientific, Auckland, New Zealand) for liquid chromatography with tandem mass spectrometry (LC–MS/MS) analysis in triplicate.

### 2.13. LC–MS/MS

LC–MS/MS was performed in a Dionex UltiMateTM 3000 RSLCnano coupled to a LTQ Orbitrap XL mass spectrometer via a nanospray ion source (Thermo Fisher Scientific, Auckland, New Zealand). Peptides were fractioned on a PepMap C18 column (3 μm, 300 Å, 75 μm × 15 cm; ThermoFisher Scientific, Auckland, New Zealand) on a 350 min gradient from 0 to 80% ACN in 0.1% FA at a constant flow rate of 300 nL/min and eluted into the Orbitrap via a PicoTip emitter (360 × 20 μm; New Objective, Littleton, MA, USA) at a voltage set to 1.8 kV through a transfer tube of 25 μm inner diameter [31,32,33]. The six most intense peptide ions from the MS scan were selected and fragmented using collision-induced dissociation (normalised collision energy, 35%; activation Q, 0.250; and activation time, 30 ms) for MS/MS scans. Dynamic exclusion was used with the following settings: repeat count, 2; repeat duration, 30 s; exclusion list size, 500; exclusion duration, 90 s [32]. The spectra was acquired using Xcalibur (version 2.1.0 SP1, Thermo Fisher Scientific, Auckland, New Zealand).

### 2.14. Protein Identification

The LC–MS/MS spectra data were searched against the Uniprot cavia porcellus protein database (25,722 sequences) using Proteome Discoverer (PD, version 2.1, Thermo Fisher Scientific) with SEQUEST HT algorithm to identify the proteins with the following settings: peptide length range 6–144, allowing 2 missed trypsin cleavages, precursor mass tolerance 10 ppm, fragment mass tolerance 0.6 Da; cysteine carboxyamidomethylation static modification (+57.021); dynamic modifications included: oxidation (+15.995 Da) at M, carbamylation (+43.066 Da) at K, acetylation (+42.011 Da) at K, deamidation (+0.984 Da) at N and Q, peptide N-terminus modification of carbamylation (+43.066 Da). The files were searched against the protein sequence database and the decoy database (Percolator node) with a false discovery rate (FDR) of 0.01, plus a relaxed FDR of 0.05. Proteins were identified with high peptide confidence and a minimum peptide number of 1 [32].

### 2.15. Label-Free Quantitation of Proteins

Label-free quantitation (LFQ) based on steracyl counts was performed using Scaffold (version 4.3.2, Proteome Software Inc., Portland, OR, USA). The PD outputs of the three LC–MS/MS technical replicates for each of the eight groups, (D21 weanling CD males, FD males, CD females and FD females) and (4 M adult CD males, FD males, CD females and FD females), were uploaded and combined in Scaffold and the total number of MS/MS spectra was calculated [31]. The built-in spectral count normalisation function and Fisher’s exact test, statistical test used for small sample sizes [31,32], were used to calculate the ratios of protein abundances and *p*-values between the control and fructose groups. Proteins were considered positive identifications and quantified when detected with 95% probability (Protein FDR ≤ 0.1%) assigned by ProteinProphet [32,34] containing at least one peptide that was detected with 95% probability (Peptide FDR ≤ 0.6%) assigned by PeptideProphet [32,35]. Proteins were considered significantly differentially expressed if *p* ≤ 0.05 according to Fisher’s *t*-test with the Benjamini Hochberg multiple test correction [31] ≥1.5.

### 2.16. Ingenuity Pathway Analysis

Ingenuity Pathways Analysis (IPA) (QIAGEN, Redwood City, CA, USA) was used to identify biological networks by conducting a core analysis to interpret the significantly differentially expressed proteins in a biological context by identifying relationships, mechanisms, functions and their associated pathways. The differentially expressed proteins with *p* ≤ 0.05 included determination of up-regulated and down-regulated proteins and networks of the molecular interactions among proteins in association with biological functions and diseases. Canonical pathway analysis was used to determine the most significantly affected signaling pathways based on increased or decreased protein expression and directionally theorised pathway activation or inhibition based on previously published evidence.

### 2.17. Western Blot

Proteins were extracted from homogenized frozen liver (~50 mg) in radioimmunoprecipitation assay (RIPA) buffer containing protease inhibitors (leupeptin (1 ug/mL and aprotinin 1 mM) and phosphatase inhibitors (25 mM NaF, 1 mM Na_2_VO_4_). Protein concentrations were determined by Pierce Bicinchoninic acid (BCA) assay (ThermoFisher Scientific, Waltham, MA, USA) according to manufacturer’s instructions. Liver homogenate samples were then diluted using RIPA buffer and 2× Laemmli SDS-PAGE sample buffer (60 mM Tris-HCl pH 6.8; 20% glycerol; 2% SDS; 4% beta-mercaptoethanol; 0.01% bromophenol blue) and denatured by heating at 95 °C for 5 min. Proteins were separated on a 10% or 12% Bis-Tris acrylamide gels. Proteins were then electro-transferred onto Immun-Blot polyvinylidene difluoride (PVDF) membranes (Bio-Rad, Hercules, CA, USA).

Membranes were blocked for 1 h at 18 °C in Tris-buffered saline/0.1% Tween 20 (TBST) containing 5% BSA or 5% skim milk then probed with primary antibodies OxPhos Rodent WB antibody cocktail binding mitochondrial complexes I-V (7076S, Cell Signalling Technology, Danvers, MA, USA), and Recombinant anti-VDAC1/porin antibody (ab1856, Abcam, Cambridge, UK), both at final dilution 1:5000 in 5% BSA. SREBP-1c (sc366, Santa Cruz Inc., Santa Cruz, TX, USA) and FAS (sc20140, Santa Cuz Inc., TX, USA), both at final dilution 1:1000 in 5% skim milk. All membranes were incubated overnight at 4 °C. The next day membranes were rinsed in TBST and probed with secondary antibodies, Rat anti-Mouse IgG (AB_466650, Thermofisher Scientific) and Anti-rabbit HRP-linked antibody (7074S, Cell Signalling Technology, Danvers, MA, USA), both at final 1:5000 in 5% BSA. OxPhos membranes were stripped with Antibody Stripping Buffer (Gene Bio Application, Yavne, Israel) and rinsed with water prior to being blocked for VDAC primary antibody. Anti-rabbit HRP-linked antibody (7074S, Cell Signalling Technology, Danvers, MA, USA) at final dilution 1:1000 in 5% skim milk and incubated for 1 h at room temperature, rinsed in TBST, then visualized using Bio-Rad Clarity ECL Substrate and Bio-Rad ChemiDoc Imager and analysed with Bio-Rad Image Lab software Version 6.0.1. All Western blots were run in duplicate.

### 2.18. Statistical Analysis

Statistical analyses were performed using IBM SPSS Statistics 25 software (IBM, Armonk, NY, USA). All data were graphed using Prism 8 software (GraphPad Software Inc., San Diego, CA, USA). Offspring biometric, biochemistry, histology and western blotting data were analysed using two-way ANOVA with diet and sex as factors. Repeated measures GEE for offspring weight gain with diet, sex and time as factors and Bonferroni post-hoc test was performed where indicated for multiple comparisons testing between groups. Offspring OGTT glucose, insulin and Matsuda–DeFronzo Insulin Sensitivity Index (M–ISI) were analysed by an independent t-test with Leven’s robust test for equality of variance. Differences between groups were considered significant at *p* < 0.05. All data were presented as mean ± SEM unless stated otherwise.

## 3. Results

### 3.1. Effects of Maternal Fructose Intake on Adult Offspring Growth and Weight Gain

CD and FD male offspring were observed to have a significant weight gain compared to CD and FD female offspring, an effect of sex (*p* ≤ 0.0001) (Figure 1). Significant differences in male and female organ weights and organ/body weight ratios were observed in FD male and female liver (*p* ≤ 0.0001) (Appendix A).

### 3.2. Effects of Excess Maternal Fructose Intake on Adult DXA Body Composition

Excess maternal fructose intake was observed to have significant effects on adult offspring bone mineral content (BMC) and bone mineral density (BMD). FD males and females had significantly decreased BMC compared to CD offspring in response to their diets (CD males, 24.28 ± 0.46 vs. FD males, 22.96 ± 0.46 vs. CD females, 19.6 ± 0.46 vs. FD females, 18.98 ± 0.46 g; *p* = 0.05). There was a significant increase in CD and FD male BMC and BMD compared to CD and FD females, an effect of sex (*p* < 0.0001 in both cases). However, FD males and females had significantly decreased BMD compared to CD offspring, an effect of diet (CD males, 0.24 ± 0.00 vs. FD males, 0.23 ± 0.00 vs. CD females, 0.21 ± 0.00 vs. FD females, 0.21 ± 0.00 g/cm^2^; *p* = 0.02). As expected, there were significant effects of sex on male and female fat mass, lean mass, total mass and body weight (Appendix A).

### 3.3. The Effects of Excess Maternal Fructose Intake on Adult OGTT Glucose and Insulin

No significant differences were observed between the CD and FD male offspring oral glucose tolerance tests (OGTT) or glucose and insulin concentrations. No significant differences were observed between CD and FD female offspring OGTT glucose and insulin concentrations. M–ISI was calculated to evaluate whole-body insulin sensitivity. No significant differences in M–ISI were observed between any of the groups (Appendix A).

### 3.4. Effects of Excess Maternal Fructose Intake on Adult Plasma Metabolites

TAG concentrations were observed to be significantly higher in FD males and females compared to CD males and females (*p* = 0.03) (Table 1). Additionally, there was an effect of sex (*p* = 0.007) with increased concentrations in CD and FD female offspring compared to CD and FD male offspring. A significant effect of sex was observed in CD and FD female offspring compared to CD and FD male offspring with increased cholesterol (*p* = 0.006) and LDL (*p* = 0.04). Fructosamine concentrations significantly increased in FD females compared to any other group or sex, an interaction effect of diet and sex (*p* = 0.05). ALP concentrations significantly decreased in FD male and female offspring compared to their CD counterparts (*p* = 0.05). No significant differences were observed between CD and FD offspring for HDL, UA, LIP, hs-CRP, ALT, GGT or leptin (Table 1).

### 3.5. Effects of Excess Maternal Fructose Intake on Adult Whole Blood Fatty Acid Composition

Excess maternal fructose intake was observed to have significant effects on adult offspring free fatty acid (FFA) composition (Table 2). Of the 33 whole-blood FFAs analysed, a significant dietary effect was observed in increased palmitoleic acid (*p* = 0.05), total omega-7 (*p* = 0.05), alpha-linolenic acid (ALA) (*p* = 0.05), total omega-3 (*p* = 0.04) and total monosaturated FFA (*p* = 0.05) levels in males and females of FD dams at 4 M (Figure 2; Table 2).

From the remaining 28 FFAs analysed, there were significant effects in eicosapentaenoic acid, linoleic acid (LA), myristic acid, arachidic acid, total trans FFA, gondoic acid, gamma-linolenic acid (GLA) and dihomo-γ-linolenic acid (DGLA) at 4 M. The remaining FFAs were not significant between CD and FD offspring. Eicosapentaenoic acid level significantly increased in CD and FD females compared to their CD and FD counterparts, an overall effect of sex (*p* = 0.05). Linoleic acid level was observed to be significantly higher in FD females compared to any other group, an interaction effect of diet and sex (*p* = 0.03). Myristic acid concentration significantly decreased in FD male and female offspring compared to CD male and female offspring, an effect of diet (*p* = 0.01). Arachidic acid significantly increased in FD males compared to CD males and FD females were significantly decreased compared to any other group, an interaction effect (*p* = 0.001). Total trans FFA significantly increased in FD male and female offspring compared to CD males and females, an effect of diet (*p* = 0.04). Gondoic acid significantly increased in FD males compared to any other group but significantly decreased in FD females compared to CD females: interaction effect (*p* = 0.002). Gamma-linolenic acid significantly decreased in FD males and females compared to CD males and females, an effect of diet (*p* = 0.003). Dihomo-γ-linolenic acid significantly increased in CD female and FD female offspring compared to CD male and FD male offspring, an effect of sex (*p* = 0.05) (Table 2).

### 3.6. Effects of Excess Maternal Fructose Intake on Offspring Hepatic Lipid Deposition

No significant differences were observed between CD and FD offspring total % hepatic lipid deposition or total % lipid deposition to liver weight ratio in weanling or adult groups (Appendix A).

### 3.7. Adult Offspring Visceral Fat Morphology

No significant differences were observed between CD and FD offspring visceral fat cell morphology (Appendix A).

### 3.8. Proteomics and Ingenuity Pathways Analysis

#### 3.8.1. Weanling Males

LFQ proteomics identified 313 significantly differentially expressed hepatic proteins in weanling (d21) FD male offspring (*n* = 5) compared to weanling CD male offspring (*n* = 5). Excess maternal fructose intake significantly increased the expression of 69 hepatic proteins in weanling FD males compared to weanling CD males. Canonical pathway analysis with IPA of the 69 proteins revealed the upregulation of oxidative phosphorylation (−log_10_
*p* = 1.9) and mitochondrial dysfunction (−log_10_
*p* = 3.83) (Appendix A) as well as NRF2-mediated oxidative stress response (−log_10_
*p* = 3.82) (Appendix A), (Appendix A).

#### 3.8.2. Weanling Females

Three hundred and thirteen significantly differentially expressed hepatic proteins were identified by total hepatic proteome analysis in weanling (D21) female offspring compared to weanling CD females. Excess maternal fructose intake significantly increased the expression of 42 hepatic proteins in weanling FD females compared to weanling CD females, and these proteins were found to be responsible for the upregulation of fatty acid β-oxidation I pathways in mitochondria (−log_10_
*p* = 3.12) (Appendix A) and NRF2-mediated oxidative stress response (−log_10_
*p* = 3.2), (Appendix A).

#### 3.8.3. Adult Males

The whole hepatic proteome analysis by LFQ identified 416 significantly differently expressed proteins in adult (4 M) FD male offspring when compared to adult (4 M) CD males. Excess maternal fructose intake significantly increased the expression of 84 hepatic proteins in adult FD males compared to adult CD males, and these proteins were demonstrated to be responsible for the upregulation of the pathway of mitochondrial dysfunction (−log_10_
*p* = 1.58) (Appendix A), (Appendix A).

#### 3.8.4. Adult Females

In all, 416 significantly differentially expressed hepatic proteins were identified by total hepatic proteome analysis in adult (4 M) FD female offspring when compared to weanling CD females. Excess maternal fructose intake significantly increased the expression of 102 hepatic proteins in adult FD females compared to adult CD females, and these proteins were found to be responsible for the upregulation of the pathways of fatty acid β-oxidation III (−log_10_
*p* = 1.87), oxidative phosphorylation (−log_10_
*p* = 1.53) and mitochondrial dysfunction (−log_10_
*p* = 2.06) (Appendix A), (Appendix A).

### 3.9. Western Blot Analysis

#### 3.9.1. Effects of Excess Maternal Fructose Intake on Key Hepatic Proteins

Relative protein abundance was determined for key proteins identified by proteomic analysis and their role in lipid synthesis and metabolism, ATP production and diffusion of metabolites across the outer mitochondrial matrix. Western blot analyses were performed to determine relative protein abundance of oxidative phosphorylation (mitochondrial ETC complexes I, II, III and IV and ATP synthase), VDAC1, SREBP-1c and FAS.

#### 3.9.2. Mitochondrial ETC Complex I

No significant differences were observed in the hepatic mitochondrial ETC complex I relative protein abundance between CD and FD weanling (Figure 3A) or adult (Figure 4A) offspring.

#### 3.9.3. Mitochondrial ETC Complex II

Weanling FD male and female offspring were shown to have significantly increased hepatic mitochondrial ETC complex II relative protein abundance compared to CD male and female offspring, an effect of diet (CD males, 0.04 ± 0.03 vs. FD males, 0.36 ± 0.03 vs. CD females, 0.04 ± 0.03 vs. FD females 0.32 ± 0.03; *p* < 0.0001) (Figure 3B). Adult FD male and female offspring were observed to have significantly increased hepatic mitochondrial complex-II relative protein abundance compared to CD male and female offspring, an effect of diet (CD males, 0.03 ± 0.00 vs. FD males, 0.08 ± 0.00 vs. CD females, 0.03 ± 0.00 vs. FD females, 0.08 ± 0.00; *p* = 0.001) (Figure 4B).

#### 3.9.4. Mitochondrial ETC Complex III

Weanling FD male and female offspring were shown to have significantly increased hepatic mitochondrial ETC complex III relative protein abundance compared to CD male and female offspring, an effect of diet (CD males, 9.80 ± 0.68 vs. FD males, 12.65 ± 0.68 vs. CD females, 7.79 ± 0.68 vs. FD females 10.54 ± 0.68; *p* = 0.04) (Figure 3C). No significant differences were observed between CD and FD adult offspring hepatic mitochondrial ETC complex III (Figure 4C).

#### 3.9.5. Mitochondrial ETC Complex IV

Weanling FD male and female offspring were shown to have significantly increased hepatic mitochondrial complex-IV relative protein abundance compared to CD male and female offspring, an effect of diet (CD males, 0.015 ± 0.00 vs. FD males, 0.024 ± 0.00 vs. CD females, 0.019 ± 0.00 vs. FD females 0.022 ± 0.00; *p* = 0.01) (Figure 3D). Adult FD male and female offspring were observed to have significantly increased hepatic mitochondrial complex-IV relative protein abundance compared to CD male and female offspring, an effect of diet (CD males, 0.01 ± 0.00 vs. FD males, 0.03 ± 0.00 vs. CD females, 0.01 ± 0.00 vs. FD females, 0.03 ± 0.00; *p* < 0.0001) (Figure 4D).

#### 3.9.6. Mitochondrial ATP Synthase

Weanling FD male and female offspring were shown to have significantly increased hepatic mitochondrial ATP synthase relative protein abundance compared to CD male and female offspring, an effect of diet (CD males, 9.44 ± 0.89 vs. FD males, 13.60 ± 0.89 vs. CD females, 4.25 ± 0.89 vs. FD females 10.75 ± 0.89; *p* < 0.0001). In addition, there was a significant increase, an effect of sex (*p* = 0.003), observed in weanling CD and FD males compared to weanling CD and FD females (Figure 3E). No significant differences were observed between adult CD and FD offspring hepatic mitochondrial ATP synthase (Figure 4F).

#### 3.9.7. Hepatic VDAC1, SREBP-1c and FAS

Weanling FD male and female offspring were shown to have significantly increased hepatic VDAC1 relative protein abundance compared to CD male and female offspring, an effect of diet (CD males, 0.09 ± 0.01 vs. FD males, 0.23 ± 0.01 vs. CD females, 0.22 ± 0.01 vs. FD females 0.25 ± 0.01; *p* <0.0001) (Figure 5A). Adult FD male and female offspring were observed to have significantly decreased hepatic VDAC1 relative protein abundance compared to CD male and female offspring, an effect of diet (CD males, 0.38 ± 0.02 vs. FD males, 0.15 ± 0.02 vs. CD females, 0.25 ± 0.02 vs. FD females, 0.14 ± 0.02; *p* < 0.0001). Additionally, adult CD and FD females were shown to have significantly decreased VDAC1 compared to CD and FD males, an effect of sex (*p* = 0.01). An interaction effect of diet and sex (*p* = 0.05) in adult FD males VDAC1 was also shown to be significantly decreased compared to any other group (Figure 6A).

Weanling FD male and female offspring were shown to have significantly decreased hepatic SREBP-1c relative protein abundance compared to CD male and female offspring, an effect of diet (CD males, 158.41 ± 10.01 vs. FD males, 71.10 ± 10.01 vs. CD females, 124.80 ± 10.01 vs. FD females 63.96 ± 10.01; *p* < 0.0001) (Figure 5B). Adult FD male and female offspring were observed to have significantly increased hepatic SREBP-1c relative protein abundance compared to CD male and female offspring, an effect of diet (CD males, 261.79 ± 52.86 vs. FD males, 764.68 ± 52.86 vs. CD females, 318.25 ± 152.86 vs. FD females, 498.61 ± 52.86; *p* < 0.0001). An interaction effect of diet and sex (*p* = 0.04) in adult FD males SREBP-1c was also shown to be significantly increased compared to any other group (Figure 6B).

Weanling FD male and female offspring were shown to have significantly increased hepatic FAS relative protein abundance compared to CD male and female offspring, an effect of diet (CD males, 21.82 ± 12.61 vs. FD males, 135.74 ± 12.61 vs. CD females, 46.17 ± 12.61 vs. FD females 80.55 ± 12.61; *p* = 0.001). In addition, there was a significant increase, an interaction effect of diet and sex (*p* = 0.03), observed in weanling FD male FAS relative protein abundance compared to any other group (Figure 5C). Adult FD male and female offspring were observed to have significantly increased hepatic FAS relative protein abundance compared to adult CD male and female offspring, an effect of diet (CD males, 0.66 ± 1.22 vs. FD males, 4.86 ± 1.22 vs. CD females, 3.46 ± 1.22 vs. FD females, 12.61 ± 1.22; *p* = 0.001). In addition, there was a significant increase in adult CD and FD female FAS compared to adult CD and FD males, an effect of sex (*p* = 0.005) (Figure 6C).

## 4. Discussion

In the current study, we investigated the effects of excess maternal fructose intake during pregnancy on the reprogramming of mitochondrial metabolism and function in weanling and young adult offspring liver. We further studied relative expression of the complete hepatic proteome in offspring livers at D21 and 4 M using label-free quantitation proteomics. Functional analysis of differentially expressed proteins showed that the up-regulated proteins increased β-oxidation, oxidative phosphorylation and mitochondrial function in male and female fructose offspring livers. Even though the dams did not have access to fructose during lactation and offspring never consuming added fructose themselves, our data show that excess maternal fructose intake, at 10% *w/v*, during in utero development permanently alters hepatic mitochondrial function, de novo lipogenesis and lipid metabolism in offspring. We showed that excess maternal fructose intake can program hepatic mitochondrial metabolism and function, which alters the underlying molecular phenotype, leading to an asymptomatic predisposition of metabolic disease in offspring during adulthood.

Although human studies showing the long-term in utero effects of excess maternal fructose intake are lacking, there is evidence in animal models to show that excess fructose during pregnancy negatively affects pregnancy outcomes [36,37,38]. In animal models, excess maternal fructose intake shows an association with hepatic de novo lipid accretion and, potentially, early onset NAFLD [2,39]. However, histological assessment of hepatic lipid content by Oil-red-O staining and hepatic triglyceride content at D21 and 4 M, showed no differences. Analysis of hepatic metabolomics revealed significant differences in specific FFAs, acyls and sphingolipids, indicating potential increases in hepatic lipid accretion. In agreement with our hypothesis, these animals are likely to display an underlying asymptomatic molecular phenotype and increased predisposition to liver dysfunction.

Palmitoleic acid is an omega-7 monounsaturated fatty acid synthesised from palmitic acid by stearoyl-CoA desaturase 1. It is the second most abundant fatty acid and is found in high concentrations in the liver and triglycerides of adipose tissue [40] and is associated with de novo lipogenesis and early-stage progression of NAFLD. However, limited information exists regarding the impact of palmitoleic acid in a DOHaD context. We observed increased palmitoleic acid and total omega-7 across all time points in offspring, suggesting a programming effect of in utero exposure to excess fructose on free fatty acid synthesis and FA oxidation and subsequent excess palmitoleic acid production. Studies have shown increases in serum palmitoleic acid represent a shift of carbon from carbohydrates to free fatty acids and visceral lipid cell lipolysis [40,41,42]. An interesting study by Okada et al. showed that increased serum palmitoleic acid was correlated with abdominal adiposity in obese children, and endogenous lipogenesis may be an important factor in the pathogenesis of obesity in children [42]. However, in the current study, no significant effects of diet on body weight, nor did we observe any differences in visceral fat cell size or number. Perhaps this may have been observed if DXA analysis had been split by body region to improve observation of any differences in regional adiposity. Bernardi et al. investigated early life stress and subsequent metabolic outcomes following dietary deficiency of omega-3 fats, which demonstrated exacerbated programmed responses in circulating palmitoleic acid during adulthood leading to insulin and leptin resistance [43]. Another recent study reported that palmitoleic acid can increase adipocyte lipolysis via PPARα-dependent mechanisms [44]. Moreover, Cruz et al. reported treatment with palmitoleic acid enhanced mitochondrial activity and increased gene expression of key catalytic enzymes in ETC complexes I, II, III and V [45]. Further studies have shown palmitoleic acid to be anti-inflammatory and may help in reducing insulin sensitivity, although the data are conflicting in the current literature [46]. In the current study, significant increases of palmitoleic acid across the lifespan in FD male and a reduction in female’s offspring may be an adaptive protective mechanism. However, the specific mechanisms responsible for the significantly elevated palmitoleic acid from birth to adulthood remain unclear. It would be interesting to observe how adult offspring would be affected by a ‘second hit’ of fructose to determine if this molecular phenotype leads to NAFLD later in life.

The role of dietary fructose in increasing de novo lipogenesis and progression of fatty liver disease has been previously postulated [47]. Our study shows up-regulation of SREBP-1c and FAS in the 4 M adult FD offspring, which may play a pathologic role in the predisposition to liver dysfunction. Increased expression of SREBP-1c in rats caused a 26-fold increase in fatty acid synthesis and significant increases in fat accumulation [48]. SREBP-1c has also been shown to increase fatty acid elongation and TAG production [49,50]. Clayton et al. used a rat model to show that genes related to hepatic lipogenesis (SREBP-1c) increased in neonate offspring (postnatal D10) of dams that consumed 10% fructose *w/v*, thereby producing a lipogenic phenotype [39]. As in the current study, increased key regulatory proteins involved in fatty acid synthesis and triglyceride production such as acetyl-CoA, SREBP-1c, FAS and stearoyl-CoA desaturase (SCD) may be early indicators of dysfunctional de novo lipogenesis.

We present data that confirms the previous findings of the programmed effects of maternal fructose intake on offspring hepatic FAS and SREBP-1c. In the livers of weanling and adult fructose offspring, we showed markers of de novo lipogenesis and increased fatty acid metabolites. Due to the differential activity of SREBP-1c over time, we consider FAS to be a preferential target of early developmental programming. Consistent increases in lipid synthesis as shown by increased FAS expression, may lead to increased conversion of the precursor palmitic acid into palmitoleic acid. Increased palmitoleic acid concentrations in both weanlings and adults, would also support this hypothesis. Increases in FAS and associated de novo lipogenic enzymes have been widely reported in fructose feeding models [51] and in humans following acute fructose feeding [47]. Studies investigating maternal fructose intake have also reported similar increases in de novo lipogenic enzymes, increased plasma and hepatic triglyceride accumulation and increased circulating fatty acids, as presented here. The precise mechanisms driving the offspring hyperlipidemia and deleterious FFA profiles are not fully understood. However, the current study demonstrated how maternal fructose can influence the developmental programming of offspring lipid synthesis and metabolism during critical windows of plasticity, which may alter an individual’s susceptibility to metabolic disease.

The data presented here showed a programming effect of fructose in male and female offspring hepatic mitochondrial metabolism, specifically in complex II and IV. We showed increased NRF2 and reduced VDAC1 protein abundance in fructose male and female weanling offspring. However, these changes in male and female weanling NRF2 pathway activation were not present during adulthood in either sex. Similarly, Cioffi et al. found that following the fructose feeding in a rat model, NRF2 increased while mitochondrial VDAC expression was reduced [52]. Significant increases observed in fatty acid production were likely to be an adaptive protective response during early life as these changes in whole blood fatty acids and NRF2 were not observed during adulthood.

Recent observations have shown that VDAC1 plays a major role in controlling metabolic function of the mitochondria for effective energy transfer and efficient ETC function. Huizing et al. showed that reduction in VDAC1 is detrimental to pyruvate and ATP production [53]. Additionally, studies have shown reduced mitochondrial VDAC1 expression causes multiple deficiencies in the ETC and ATP production [54]. We showed significantly increased hepatic VDAC1 expression was observed in male and female weanling livers. Conversely, a significant reduction in VDAC1 expression during adulthood in male and female FD groups. Increased VDAC1 abundance is likely due to rapid cell turnover and increased metabolism during neonatal development. VDAC1 was observed to be significantly greater in weanling male and female FD offspring and significantly reduced in adult FD offspring. Reduced VDAC1, in combination with consistently significant differential protein activity in ETC complexes II and IV in adult FD offspring indicates altered hepatic mitochondrial catalytic activity. Yakubu et al. reported that VDAC1 expression in sheep reaches maximum levels at 140 days of gestation and up to day 1 following birth in the lungs, kidneys and liver, but then is reduced across all tissues by 6 months of age [55]. Similar to results presented here, these findings suggest a tissue-specific, time-dependent expression of VDAC1 that may reflect increased tissue metabolic demand and functional adaptation following birth.

The underlying mechanisms of developmentally programmed increased de novo lipogenesis may be an increase in β-oxidation in combination with increased mitochondrial oxidative phosphorylation dysfunction, both of which increased in our fructose offspring. In both male and female FDs, there were consistent increases in mitochondrial oxidative phosphorylation complexes II and IV and FAS in weanlings and adults. Due to the intimately linked pathways of de novo lipogenesis, β-oxidation and oxidative phosphorylation, we faced a classic ‘chicken or egg’ scenario. Were programmed changes in FAS contributing to increased activation of the TCA cycle and β-oxidation (and subsequently increasing complex II and IV protein expression) or were the programmed changes in mitochondrial nDNA in complex II and IV driving an altered proton gradient and decrease of ATP movement across the mitochondrial matrix by VDAC1?

Our findings showed that a maternal diet providing a moderate daily caloric intake (~16.5%) of fructose from a 10% *w/v* fructose-sweetened beverage adversely influenced the developmental programming of (a) lipid synthesis and β-oxidation shown by consistent up-regulation of the β-oxidation pathway in female offspring and increased circulating FFAs in the whole blood of male and female weanlings and adults: specifically, increased FAS and products of the malonyl—palmitate—palmitoleic acid pathway and the potential for this FFA to play a role in the early stages and progression of hepatic lipid accretion and a pro-NAFLD phenotype; and (b) increased oxidative phosphorylation and mitochondrial dysfunction in male and female livers are shown by increased FAS, SREBP-1c, mitochondrial complexes II, IV and decreased VDAC1. More remarkably, all of these changes were significantly altered in fructose offspring that had never consumed any fructose. It would be interesting to observe how adult offspring would be affected by a ‘second hit’ of fructose and what the further nutritional implications of our study in aged offspring would be.

The DOHaD concept could be important in the prevention of NCDs and reducing risk of vertical transmission to the next generation; however, there was no clearly defined relationship between maternal fructose and the prevalence of increased predisposition to metabolic disease such as type-2 diabetes, metabolic syndrome and non-alcoholic fatty liver disease. Increased hepatic lipid accumulation observed in our weanlings and younger adults at 4 M, could lead to significant hepatic and systemic consequences including mitochondrial dysfunction, non-alcoholic fatty liver disease, cirrhosis, impairment of glucose metabolism, thereby contributing to increased risk of metabolic-related disease. Furthermore, aging is characterized by changes to all aspects of physiology resulting in an increased risk of disease with time. The dysregulation of glucose and lipid metabolism has a long-standing relationship with cardio-metabolic disease. Various factors that contribute to the dysregulation include both modifiable (e.g., in utero environment, obesity, insulin resistance) and non-modifiable risk factors such as age-related metabolic adaptations. Age-related changes superimposed upon an underlying molecular phenotype that is already predisposed to metabolic dysfunction could have profound effects on the later-life health of offspring. However, the multifaceted nature of fructose, lipid metabolism and the complexities of its interaction with aging make it challenging to determine the underlying pathways. Future work should aim to improve our understanding of the mechanisms that underlie the metabolic consequences in response to further nutrient specific interventions following intrauterine exposure to fructose. This would help to identify targets and treatments that could be used to mitigate the premature onset of metabolic and age-related metabolic diseases.

In conclusion, we provided evidence of programmed hepatic mitochondrial function and associated increases in lipid metabolism throughout life following maternal fructose exposure. We presented robust evidence demonstrating the detrimental effects of maternal fructose intake on offspring physiology and the asymptomatic molecular profile, predisposing offspring to hepatic metabolic dysregulation throughout life. It may be prudent for women considering pregnancy and indeed potential fathers, to limit their intake of extrinsic fructose prior to and during pregnancy and lactation. Furthermore, this data may assist in understanding how excess maternal fructose influences an offspring’s mechanistic pathways, including genes and proteins linked with β-oxidation and oxidative phosphorylation.

## Figures and Tables

**Figure 1 ijms-23-00999-f001:**
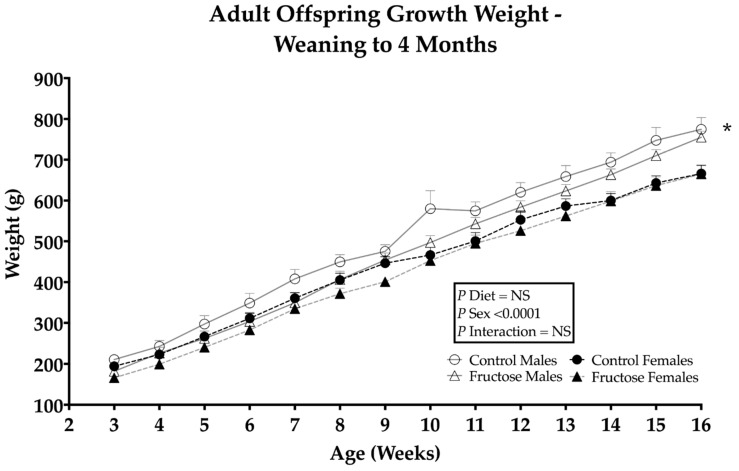
Weanling (D21) to adult (4 M) offspring weight gain from week 3 to week 16. CD males (*n* = 7); FD males (*n* = 7); control females (*n* = 7) and fructose females (*n* = 7). Significant effects were shown in the sex of offspring weight gain. All data were analysed using a 2 × 2 factorial design with repeated measures GEE, diet*sex*time*interaction included as factors (general analysis of variance) using IBM SPSS statistics 25. Data presented as group mean ± SEM. * denotes significance of *p* < 0.0001.

**Figure 2 ijms-23-00999-f002:**
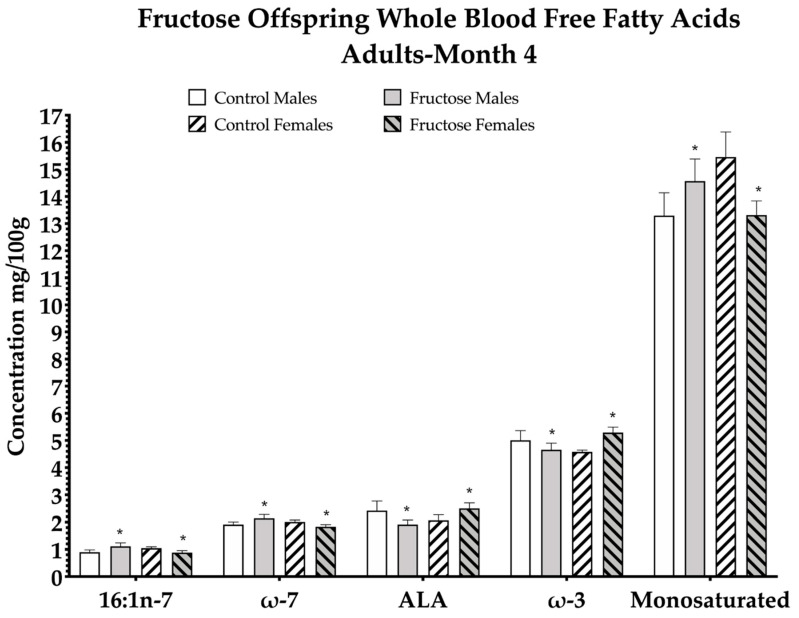
Free fatty acid (FFA) concentrations in the blood of adult (4 M) offspring. CD males (*n* = 7); FD males (*n* = 9); CD females (*n* = 7) and FD females (*n* = 6). Figure represents offspring month 4 whole blood FFA. Palmitoleic acid (16:1n-7), total omega-7 (**ω**-7), alpha linolenic acid (ALA), total omega-3 (**ω**-7). All data were analysed using a 2 × 2 factorial design with diet*sex*interaction included as factors (general analysis of variance) using IBM SPSS statistics 25. Data presented as group mean ± SEM. * denotes significance of *p* < 0.05.

**Figure 3 ijms-23-00999-f003:**
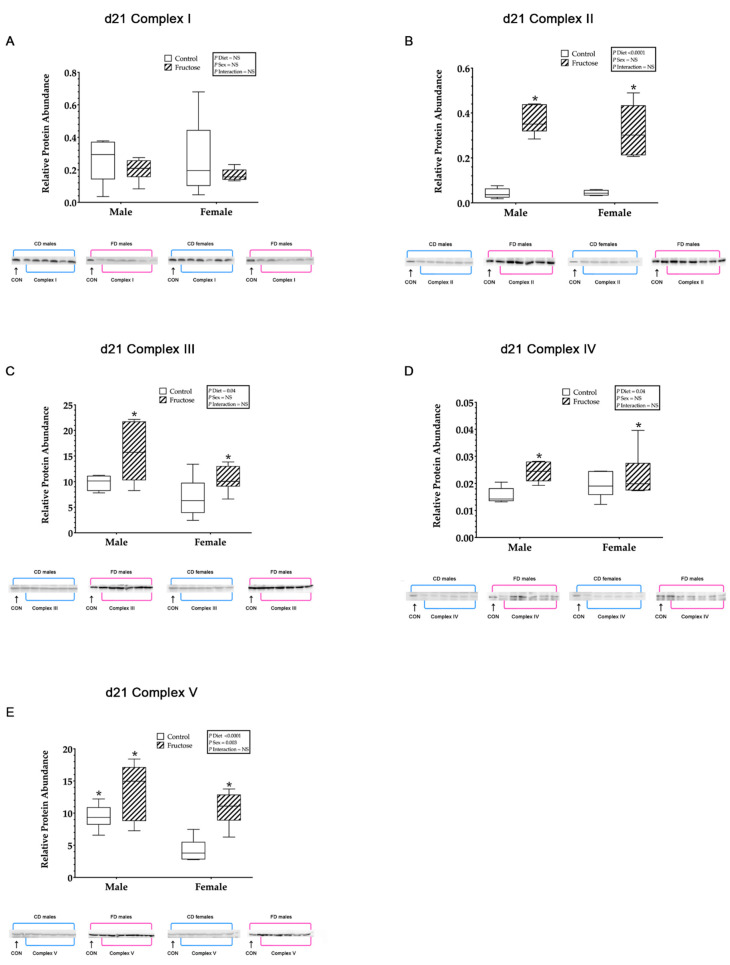
(**A**–**D**) Weanling and adult offspring hepatic mitochondrial electron transport and oxidative phosphorylation complexes I, II, III, IV and ATP synthase. Weanling offspring control males (*n* = 6); fructose males (*n* = 6); control females (*n* = 6) and fructose females (*n* = 6). (**A**) Represents weanling offspring hepatic mitochondrial complex-I relative protein abundance and Western blot image at D21. (**B**) Represents weanling offspring hepatic mitochondrial complex-II relative protein abundance and Western blot image at D21. (**C**) Represents weanling offspring hepatic mitochondrial complex-III relative protein abundance and Western blot image at D21. (**D**) Represents weanling offspring hepatic mitochondrial complex-IV relative protein abundance and Western blot image at D21. (**E**) Represents weanling offspring hepatic mitochondrial complex-V relative protein abundance and Western blot image at D21. All data were analysed using a 2 × 2 factorial design with diet*sex*interaction include as factors (general analysis of variance) using IBM SPSS statistics 25. Data presented as group mean ± SEM. * denotes significance of *p* < 0.05.

**Figure 4 ijms-23-00999-f004:**
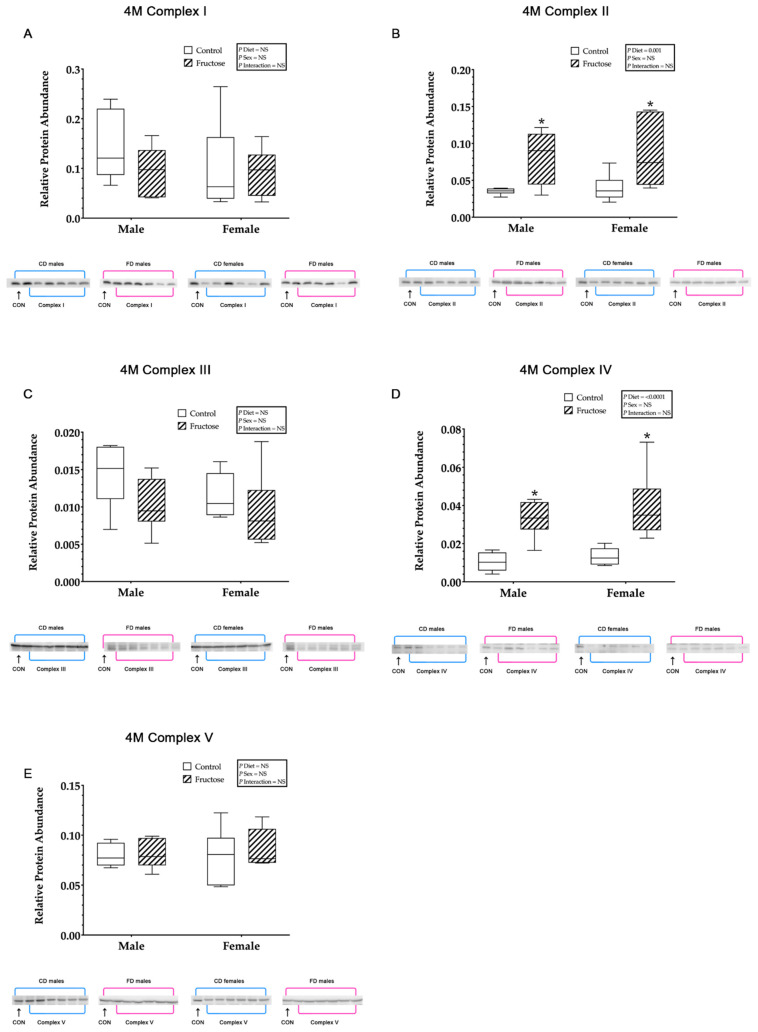
(**A**–**D**) Adult offspring control males (*n* = 6); fructose males (*n* = 6); control females (*n* = 5) and fructose females (*n* = 6). (**A**) Represents adult offspring hepatic mitochondrial complex-I relative protein abundance and Western blot image at 4 M. (**B**) Represents adult offspring hepatic mitochondrial complex-II relative protein abundance and Western blot image at 4 M. (**C**) Represents adult offspring hepatic mitochondrial complex-III relative protein abundance and Western blot image at 4 M. (**D**) Represents adult offspring hepatic mitochondrial complex-IV relative protein abundance and Western blot image at 4 M. (**E**) Represents adult offspring hepatic mitochondrial ATP synthase relative protein abundance and Western blot image at 4 M. All data were analysed using a 2 × 2 factorial design with diet*sex*interaction include as factors (general analysis of variance) using IBM SPSS statistics 25. Data presented as group mean ± SEM. * denotes significance of *p* < 0.05.

**Figure 5 ijms-23-00999-f005:**
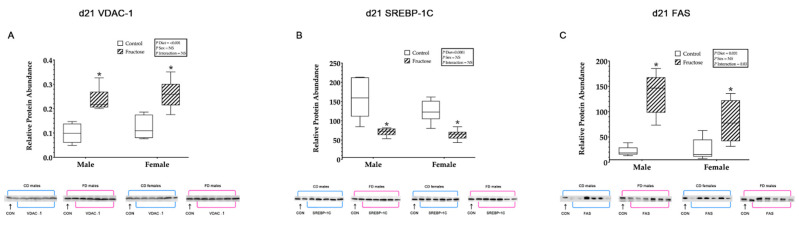
(**A**–**C**) Weanling offspring hepatic proteins. Weanling offspring control males (*n* = 6); fructose males (*n* = 6); control females (*n* = 6) and fructose females (*n* = 6). (**A**) Represents weanling offspring hepatic VDAC1 relative protein abundance and Western blot image at D21. (**B**) Represents weanling offspring hepatic SREBP-1c relative protein abundance and Western blot image at D21. (**C**) Represents weanling offspring hepatic FAS relative protein abundance and Western blot image at D21. All data were analysed using a 2 × 2 factorial design with diet*sex*interaction include as factors (general analysis of variance) using IBM SPSS statistics 25. Data presented as group mean ± SEM. * denotes significance of *p* < 0.05.

**Figure 6 ijms-23-00999-f006:**
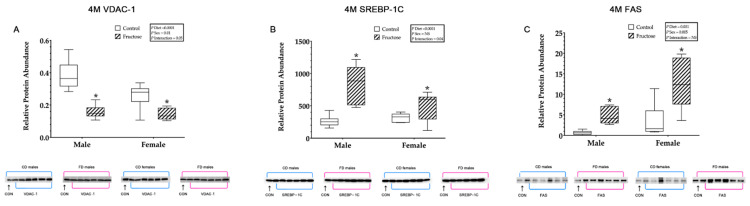
(**A**–**C**) Adult offspring hepatic proteins. Adult offspring control males (*n* = 6); fructose males (*n* = 6); control females (*n* = 6) and fructose females (*n* = 6). (**A**) Represents adult offspring hepatic VDAC1 relative protein abundance and Western blot image at 4 M. (**B**) Represents adult offspring hepatic SREBP-1c relative protein abundance and Western blot image at 4 M. (**C**) Represents adult offspring hepatic FAS relative protein abundance and Western blot image at 4 M. All data were analysed using a 2 × 2 factorial design with diet*sex*interaction include as factors (general analysis of variance) using IBM SPSS statistics 25. Data presented as group mean ± SEM. * denotes significance of *p* < 0.05.

**Table 1 ijms-23-00999-t001:** Adult offspring plasma metabolites at month 4.

Adult Offspring Plasma Metabolites	Sex	Month 4	
		Control	Fructose
TAG (mmol/L)	Male	0.55 ± 0.05	0.62 ± 0.05 *
	Female	0.69 ± 0.05 *	1.05 ± 0.05 **
CHOL (mmol/L)	Male	0.72 ± 0.05	0.98 ± 0.05
	Female	0.82 ± 0.05 *	1.12 ± 0.05 *
LDL (mmol/L)	Male	0.67 ± 0.05	0.71 ± 0.05
	Female	0.88 ± 0.05 *	0.96 ± 0.05 *
FRUC (umol/L)	Male	240.16 ± 6.57	222.57 ± 6.57
	Female	239.00 ± 6.57	270.80 ± 6.57 *
ALP (U/L)	Male	13.33 ± 1.37	4.88 ± 1.37 *
	Female	5.16 ± 1.37	3.50 ± 1.37 *
HDL (mmol/L)	Male	0.03 ± 0.00	0.41 ± 0.00
	Female	0.03 ± 0.00	0.05 ± 0.00
UA (µmol/L	Male	16.33 ± 2.18	24.88 ± 2.18
	Female	21.50 ± 2.18	26.83 ± 2.18
LIP (U/L)	Male	6.20 ± 0.81	10.18 ± 0.81
	Female	6.88 ± 0.81	07.31 ± 0.81
hs-CRP (mg/L)	Male	0.10 ± 0.01	0.09 ± 0.01
	Female	0.14 ± 0.01	0.09 ± 0.01
ALT (U/L)	Male	72.82 ± 9.13	99.33 ± 9.13
	Female	66.50 ± 9.13	91.51 ± 9.13
GGT (U/L)	Male	171.80 ± 15.93	168.85 ± 15.93
	Female	89.40 ± 15.93	156.75 ± 15.93
Leptin (ng/nL)	Male	4.85 ± 0.41	3.98 ± 0.41
	Female	3.93 ± 0.41	4.34 ± 0.41

CD males (*n* = 7); FD males (*n* = 9); CD females (*n* = 7) and FD females (*n* = 6). All data were analysed using a 2 × 2 factorial design with diet*sex*interaction included as factors (general analysis of variance) using IBM SPSS statistics 25. Data presented as group mean ± SEM. * denotes significance of *p* < 0.05; ** denotes significance of *p* < 0.0001.

**Table 2 ijms-23-00999-t002:** Free fatty acid concentrations in fructose adult offspring whole blood at month 4.

Adult Offspring Whole Blood FFA (mg/100 g)	Sex	Month 4	
		Control	Fructose
Palmitoleic acid (16:1n-7)	Male	0.89 ± 0.04	1.11 ± 0.04 *
	Female	1.04 ± 0.04	0.88 ± 0.04 *
Total Omega-7	Male	1.90 ± 0.05	2.14 ± 0.05 *
	Female	2.00 ± 0.05	1.83 ± 0.05 *
ALA (18:3n-3)	Male	2.41 ± 0.11	1.91 ± 0.11 *
	Female	2.06 ± 0.11	2.50 ± 0.11 *
Total Omega-3	Male	5.00 ± 0.12	4.66 ± 0.12 *
	Female	4.58 ± 0.12	5.28 ± 0.12 *
Total Monosaturated fats	Male	13.28 ± 0.42	14.57 ± 0.42 *
	Female	15.45 ± 0.42	13.31 ± 0.42 *
EPA (20:5n-3)	Male	0.14 ± 0.01	0.15 ± 0.01
	Female	0.22 ± 0.01 *	0.20 ± 0.01 *
LA (18:2n-6)	Male	19.86 ± 0.40	19.31 ± 0.40
	Female	18.05 ± 0.40 *	21.04 ± 0.40 *
Total Saturates	Male	48.09 ± 0.32 *	48.52 ± 0.32 *
	Female	46.92 ± 0.32	46.65 ± 0.32
Myristic acid (14:0)	Male	1.04 ± 0.04	0.17 ± 0.04 *
	Female	0.95 ± 0.04	0.80 ± 0.04 *
Arachidic acid (20:0)	Male	0.68 ± 0.19	0.76 ± 0.19 *
	Female	0.79 ± 0.19	0.62 ± 0.19 *
Total Trans Fatty acids	Male	0.65 ± 0.09	1.01 ± 0.09 *
	Female	0.68 ± 0.09	1.15 ± 0.09 *
Gondoic Acid (20:1n-9)	Male	0.48 ± 0.08	1.14 ± 0.08 *
	Female	0.83 ± 0.08	0.52 ± 0.08 *
GLA (18:3n-6)	Male	0.22 ± 0.01	0.13 ± 0.01 *
	Female	0.18 ± 0.01	0.13 ± 0.01 *
DGLA (20:3n-6)	Male	0.36 ± 0.01	0.37 ± 0.01
	Female	0.43 ± 0.01 *	0.43 ± 0.01 *
Margaric acid (17:0)	Male	1.45 ± 0.02	1.45 ± 0.02
	Female	1.36 ± 0.02	1.37 ± 0.02
DHA (22:6n-3)	Male	0.45 ± 0.02	0.54 ± 0.02
	Female	0.56 ± 0.02	0.56 ± 0.02
Pentadecanoic acid (15:0)	Male	0.84 ± 0.04	0.90 ± 0.04
	Female	0.83 ± 0.04	0.69 ± 0.04
dma16:0	Male	0.40 ± 0.06	0.72 ± 0.06
	Female	0.50 ± 0.06	0.44 ± 0.06
Total Omega-6	Male	31.32 ± 0.49	31.07 ± 0.49
	Female	30.83 ± 0.49	33.58 ± 0.49
Vaccenic acid (t18:1n-7)	Male	0.28 ± 0.01	0.36 ± 0.01
	Female	0.28 ± 0.01	0.31 ± 0.01
Palmitic acid (16:0)	Male	15.50 ± 0.19	15.26 ± 0.19
	Female	15.20 ± 0.19	15.64 ± 0.19
Elaidic acid (t18:1n-9)	Male	0.21 ± 0.01	0.19 ± 0.01
	Female	0.24 ± 0.01	0.20 ± 0.01
dma18:0	Male	0.42 ± 0.08	0.85 ± 0.08
	Female	0.45 ± 0.08	0.54 ± 0.08
Behenic acid (22:0)	Male	0.94 ± 0.04	0.94 ± 0.04
	Female	1.04 ± 0.04	0.92 ± 0.04
Adrenic acid (22:4n-6) + Docosatrienoate acid (22:3n-3)	Male	2.32 ± 0.05	2.27 ± 0.05
AA (20:4n-6)	Male	8.62 ± 0.15	8.47 ± 0.15
	Female	8.57 ± 0.15	8.97 ± 0.15
DPA (22:5n-3)	Male	2.15 ± 0.06	1.91 ± 0.06
	Female	1.97 ± 0.06	1.97 ± 0.06
Cis-Vaccenic acid (18:1n-7)	Male	0.97 ± 0.01	0.98 ± 0.01
	Female	0.97 ± 0.01	0.97 ± 0.01
Eicosadienoic acid (20:2n-6)	Male	0.77 ± 0.02	0.74 ± 0.02
	Female	0.66 ± 0.02	0.71 ± 0.02
Lignoceric acid (24:0)	Male	1.65 ± 0.06	1.67 ± 0.06
	Female	1.79 ± 0.06	1.77 ± 0.06
Stearic acid (18:0)	Male	24.75 ± 0.32	25.33 ± 0.32
	Female	23.47 ± 0.32	24.17 ± 0.32
Oleic acid (18:1n-9)	Male	9.59 ± 0.18	9.77 ± 0.18
	Female	10.16 ± 0.18	10.40 ± 0.18
Total Omega-9	Male	10.42 ± 0.19	10.94 ± 0.19
	Female	11.46 ± 0.19	10.95 ± 0.19

CD males (*n* = 7); FD males (*n* = 9); CD females (*n* = 7) and FD females (*n* = 6). All data were analysed using a 2 × 2 factorial design with diet*sex*interaction included as factors (general analysis of variance) using IBM SPSS statistics 25. Data presented as group mean ± SEM. * denotes significance of *p* < 0.05; * denotes significance of *p* < 0.0001.

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
