# Peer review of "Maternal Fructose Intake Causes Developmental Reprogramming of Hepatic Mitochondrial Catalytic Activity and Lipid Metabolism in Weanling and Young Adult Offspring"

_ijms, 2022, doi:10.3390/ijms23020999_

Round 1
Reviewer 1 Report
- Results: Figs 3, 4, 6, 7, and 8 are unreadable or hardly readable (Fig.9) in their current presentation and need to be increased to the size of whole page. It is difficult to interpret the data presented on these figures. Loading controls are missing for Western blots in Figs 8 and 9.
Data in Supplemental Table 2 came first in the results, so it is logical to present these data as Suppl. Table 2.
- Discussion: The manuscript will be improved if authors add more about nutritional implications of their study.
- There are multiple typos, including missing letters, interrupted sentences, etc all over the manuscript.
Author Response
Response to reviewers
Dear Editor and Reviewers,
On behalf of myself and all co-authors, I would like to thank the reviewers for the positive and constructive feedback on our manuscript and believe the manuscript has been strengthened by these suggestions and revisions. We have included the changes to text and reformatted and reorganised the figures to aid clarity upon journal formatting. We have uploaded a tracked changes copy of the manuscript and a ‘clean’ copy of the amended manuscript with the reformatted figures.
Reviewer 1
- A) Results: Figs 3, 4, 6, 7, and 8 are unreadable or hardly readable (Fig.9) in their current presentation and need to be increased to the size of whole page. It is difficult to interpret the data presented on these figures.
The authors agree with the reviewers. The original submission all files mentioned above were indeed were full page. However, journal formatting and resizing of the images have made them difficult to read. Due to the number of pathway figures, we have altered the order in which the pathway figures appear. We have removed from the main body of text and placed as supplementary file. We have now also split figures 8 and 9 into day21 and a separate file for 4 month data to aid clarity of the figures.
- B) Loading controls are missing for Western blots in Figs 8 and 9.
We have reformatted the figures to include the loaded control and the protein of interest.
- Data in Supplemental Table 2 came first in the results, so it is logical to present these data as Suppl. Table 2.
Apologies for this oversight. We would agree and have changed supplementary table 2 to supplementary table 1 and the orginal supplementary table 1 to supp. Table 2.
- Discussion: The manuscript will be improved if authors add more about nutritional implications of their study.
We have included more information relating to the nutritional implications of our study, as well as, inclusion of the impact of age-related metabolism which may also have increased impact following fetal exposure to fructose.
- There are multiple typos, including missing letters, interrupted sentences, etc all over the manuscript.
Apologies. The manuscript has been checked and all or any typo’s have been amended. Changes to text to benefit the flow of the text have also been included.
Reviewer 2.
I have red with extreme pleasure the manuscript form LaRae Smith and collaborators, illustrating the effects of maternal fructose intake on hepatic lipid metabolism in the offsprings. I have only minor suggestions to improve the quality of the manuscript:
- I think that the project involving laboratory animals has a code and a date of approval, please add these detail in the text.
The reviewer is correct and the animal ethics approval code has now been added to the methods.
- the Figures have low resolution, please improve.
Agreed. Please see reviewer #1, comment 1. All necessary changes to the figures in question have been included in the revised manuscript.
- the last time point is 4 months. The authors should add information about the dysregulation of lipid metabolism associated to aging, and correlate these literature data with what they found.
The authors have included some additional text on the impact of age-related metabolism which may also have increased impact following fetal exposure to fructose has been added to the discussion.
Reviewer 2 Report
I have red with extreme pleasure the manuscript form LaRae Smith and collaborators, illustrating the effects of maternal fructose intake on hepatic lipid metabolism in the offsprings. I have only minor suggestions to improve the quality of the manuscript:
- I think that the project involving laboratory animals has a code and a date of approval, please add these detail in the text
- the Figures have low resolution, please improve
- the last time point is 4 months. The authors should add information about the dysregulation of lipid metabolism associated to aging, and correlate these literature data with what they found.
Author Response

(The authors gave the same response as above.)

Round 2
Reviewer 1 Report
Authors should remove a duplicated small old version of Fig.3 from page 15.